# The radiation protection behavior of medical workers: A scoping review protocol

Xun Liu[1], Yaqing Liu[1], Pengyan Xiong[1], Sixuan Guo[1], Lei Zhang[1,2], Li Liao[1] *

**1** School of Nursing, University of South China, Hengyang, Hunan Province, China, **2** The First Affiliated Hospital of the University of South China, Hengyang, Hunan Province, China

☯ These authors contributed equally to this work.
* 254251558@qq.com

**Data Availability Statement:** No datasets were generated or analysed during the current study. All relevant data from this study will be made available upon study completion.

## Abstract

### Introduction

Radiation exposure in medical settings stands as the primary source of artificial radiation, compounded by the yearly rise in healthcare worker numbers. Ensuring radiation protection is crucial for safeguarding their occupational health. Nevertheless, existing studies on radiation protection behavior exhibit considerable heterogeneity due to various factors.

### Objective

This scoping review aims to explore the current status of research on radiation protection behavior and identify research gaps, intending to guide future research directions.

### Methods and analysis

The scoping review will follow the Arksey and O'Malley framework and the Joanna Briggs Institute methodology. A systematic search will be conducted across English databases including PubMed, Web of Science, Embase, and Medline, as well as Chinese databases such as CNKI, Wanfang, VIP, and China Biomedical Literature Database. Two independent reviewers will screen the studies based on predefined eligibility criteria and extract the data. Any disagreements will be resolved through discussion by a third reviewer. The review will be reported according to the Preferred Reporting Items for Systematic Reviews and Meta-Analysis extension for Scoping Reviews.

### Strengths and limitations of this study

A stakeholder consultation will provide an opportunity to validate the findings and address any potential gaps in the article.

In this scoping review, all types of studies will be considered.

The effectiveness of the methodological quality of the included studies will not be reported, which may lead to some studies of poor quality being included.

Only studies published in English or Chinese after 2010 will be considered in this review, potentially leading to the omission of relevant papers.

**Funding:** Hunan Province Wisdom Ucare Innovation and Entrepreneurship Education Center (Li Liao, No.2018380).The funders will not have a role in study design, data collection and analysis, decision to publish, or preparation of the manuscript. No additional external funding was received for this study.

**Competing interests:** The authors have declared that no competing interests exist.

## Introduction

Roentgen discovered X-rays using vacuum tubes, and the electromagnetic spectrum has since been extensively utilized in healthcare, industry, science, and various other domains [1]. Particularly in the medical field, according to a report by the United Nations Scientific Committee on the Effects of Atomic Radiation (UNSCEAR), medical exposure has emerged as the primary artificial source of ionizing radiation [2]. In China alone, by the year 2023, approximately 4 million individuals may have been exposed to ionizing radiation in their occupational settings [3,4], with medical workers (MWs) accounting for one-eighth of this population [5,6] Meanwhile, the number of MWs is steadily increasing annually.

Radiation exposure has emerged as a significant occupational hazard for medical personnel engaged in radiological work. Working in a prolonged low-dose ionizing radiation environment poses heightened risks of radiation-induced diseases, including cataracts, fertility disorders, tumors, chromosomal and cellular aberrations, immune disorders, and osteoarthropathies [7–11]. Hence, to mitigate the adverse effects of radiation, the International Radiation Protection Association (IRPA) has formulated numerous guidelines aimed at limiting the radiation dose received by medical workers, subject to periodic review [12].

Radiation protection can be classified into two main categories based on the target of protection: patients and medical workers (MWs) [13] This paper primarily focuses on the implementation of radiation protection measures by medical personnel to mitigate occupational radiation exposure, which is crucial for their occupational health. The radiation protection of MWs primarily involves shielding against external radiation during diagnosis and treatment in hospital or clinical settings [14]. Generally, three methods are employed for this type of protection: time, distance, and shielding [15]. Shielding involves wearing lead-containing equipment such as lead aprons, thyroid collars, lead glasses, and lead gloves, considered the most effective way to mitigate ionizing radiation exposure. Personal protection with all of the aforementioned equipment can reduce exposure by up to 90% [16]. While equipment and other methods are beneficial, radiation protection behaviors (RPB) should be adopted "proactively" rather than "passively".

To enhance the level of Radiation Protection Behaviors (RPB) among medical personnel and thereby promote their occupational health, it is essential to comprehensively understand the current state of research. Although there exists a systematic review reporting on radiation protection KAP among healthcare workers, encompassing Knowledge, Attitude, and Practice holistically, the analysis of practical outcomes remains incomplete [17]. Moreover, the study of RPB exhibits considerable variability due to factors such as regional disparities, research methodologies, assessment tools, and other influencing variables [18–20]. Therefore, focusing on RPB involves standardizing and systematically searching and screening the literature, elucidating the scope and breadth of the research, summarizing the findings, and identifying any research limitations.

## Objectives

This scoping review aims to examine the current research status of Radiation Protection Behaviors (RPB) among medical workers (MWs), including aspects such as the definition of radiation protection behavior, current practices of radiation protection in medical settings, and the tools used for measurement. To identify research gaps and guide future research directions.

## Methods

The forthcoming scoping review will follow the Arksey and O'Malley [21] framework and the Joanna Briggs Institute methodology for scoping reviews [22], encompassing six steps: (1)

identifying the research question; (2) identifying relevant studies; (3) study selection; (4) charting the data; (5) collating, summarizing and reporting the results and (6) consultation. This protocol has been registered through the Open Science Framework (http://osf.io/2vxkz).

## Step1: Identifying the research question

This scoping review aims to assess the current research status of radiation protection behavior. Consistent with this objective, the formulated research questions are as follows:

1. What is the precise definition of radiation protection behavior?

2. How do medical workers implement radiation protection measures?

3. What factors influence the radiation protection behavior of medical personnel?

4. Currently, what tools are utilized to assess radiation protection behavior among medical workers?

5. What behavioral interventions for radiation protection are currently in use?

## Step2: Identifying relevant studies

We will conduct a systematic search across several electronic databases, including four English databases: PubMed, Web of Science, Embase, and Medline, and four Chinese databases: CNKI, Wanfang, VIP, and China Biomedical Literature Database. Keyword search terms containing both "medical workers" and "radiation protection behavior", along with other relevant subject terms and free text terms, will be utilized to identify relevant studies. The search process will commence with PubMed and CNKI's databases. Subsequently, we will analyze the titles and abstracts of retrieved papers to refine the search strategy. In the second phase, all identified keywords and index terms will be applied across all databases. Lastly, to identify additional sources, we will screen reference lists, forward citations, and gray literature associated with studies included in this scoping review. Data screening, selection, and extraction will be independently conducted by two researchers. The initial search strategy for PubMed is shown in S1 Table.

## Step3: Study selection

After the search, all identified records will be uploaded to the reference management software Zotero. Duplicate studies will be identified and removed. The study selection process will entail a two-step screening approach, involving initial screening of titles and abstracts, followed by a full-text review. In both stages, two independent reviewers (XL and YL) will assess articles against the eligibility criteria. The flow chart of the selection of articles for review is depicted in Fig 1. Any disagreements will be resolved through discussion, and if consensus cannot be reached, a third reviewer (PX) will be consulted. Eligible studies will be selected based on the following criteria:

Inclusion:1. Studies written in English and Chinese will be included.

2. The study subjects are medical workers, including doctors, nurses, and technicians.

3. The search time frame for the database is from January 1, 2010, to December 31, 2023.

4. The article focuses on the radiation protection behavior of medical personnel.

5. There are no limitations on the type of study.

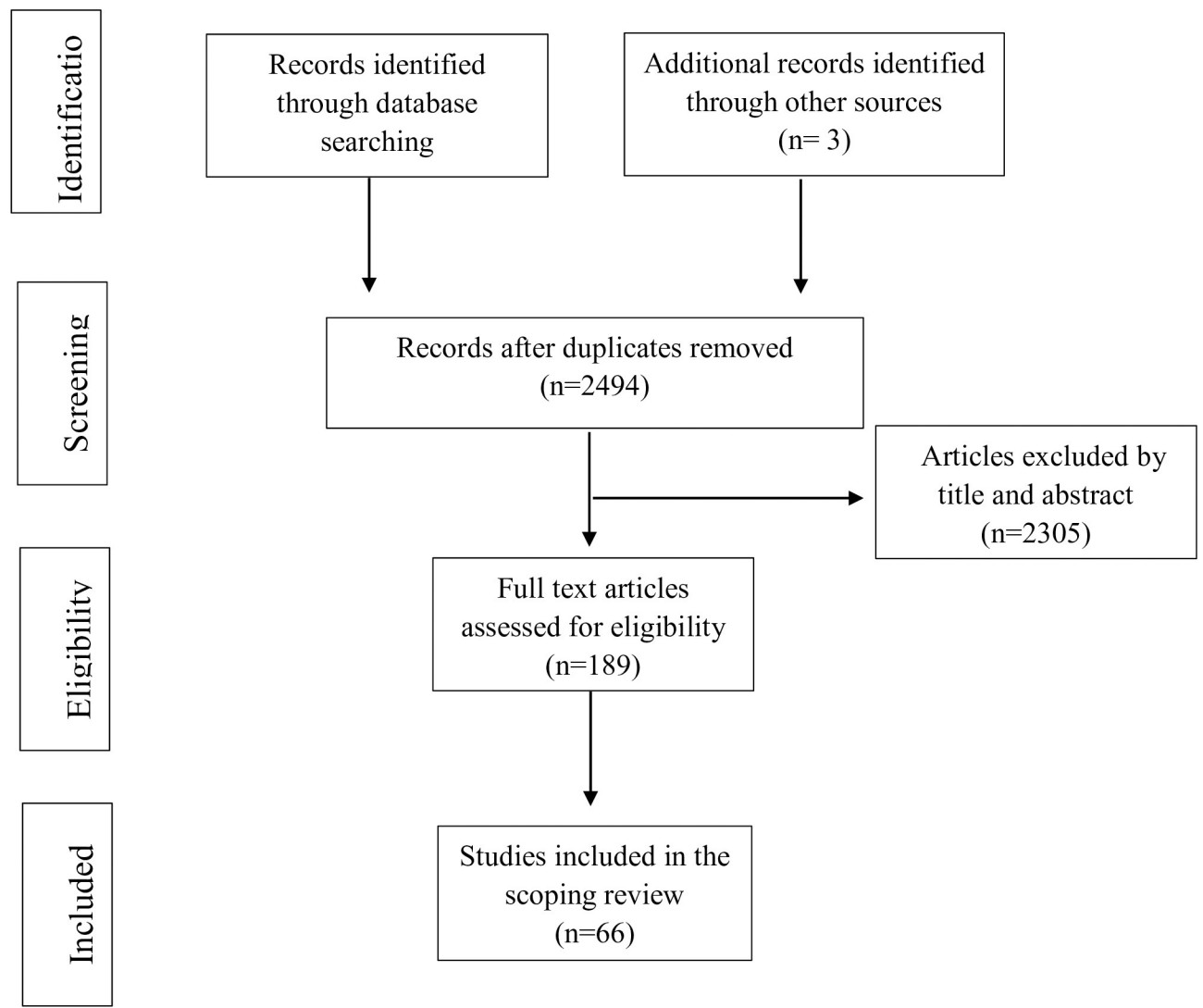

**Fig 1. Flow chart of selection of articles for review.**

Exclusion:1. Incomplete articles or articles with unavailable full texts.

2. Articles reporting study protocols.

3. Opinion pieces, viewpoints and conceptual frameworks, conference abstracts.

## Step4: Charting the data

Two independent reviewers (XL and YL) will extract relevant data from all included studies in the scoping review. A structured data recording form developed by the reviewers will be utilized, and the information will be recorded on Microsoft Excel. The extracted data will include details such as title, author names, year of publication, type of study, country of study, study object, number of participants, assessment tools, influence factors, and intervention programs. Following the extraction of data from each study, the draft data extraction tool will undergo

piloting and necessary revisions. Any disagreements between reviewers will be resolved through discussion, and a third reviewer will serve as an arbiter if consensus cannot be reached. In cases of missing or incomplete data, we will reach out to the study authors.

### Step5: Collating, summarizing, and reporting the results

We will present an overview of the data extracted from the included papers in a diagrammatic or tabular format to characterize and summarize the results. Consistent with the objective of this study, we will provide an overview of the target participants, the status of radiation protection behavior, the types of studies included, and the context of each study. A narrative summary will accompany the tabulated and/or charted results, explaining how the findings align with the objectives and aims of the scoping review. Findings will be reported following the guidelines outlined in the "Preferred Reporting Items for Systematic Reviews and Meta-Analysis: Extension for Scoping Reviews" checklist.

### Step6: Consultation

A stakeholder consultation is planned to validate the conclusions of the review, introduce new perspectives, and identify areas for future research. The stakeholders will include experienced researchers in the fields of radiation protection and medical occupational health. The consultation will involve presenting our study and findings to a panel of experts in radiation protection, followed by collating their feedback. This feedback will be incorporated into the final presentation of our article.

## Discussion

This scoping review aims to analyze the current research landscape, identifying both strengths and limitations and proposing suggestions for further investigation in this field. The strength of this research is its standardized search methods, stringent inclusion criteria, and data charting methods, which provide a comprehensive and clear understanding of this field.

There is no discernible consensus regarding the precise definition of radiation protection behaviors among medical workers. The studies in question exhibit considerable variation in terms of the specific behaviors they assess to radiation protection [23–26]. However, the measurement of RPB among medical personnel primarily encompasses radiation protection training, dosimeter utilization, the application of radiation protection equipment, and the performance of health checkups. Because the specific RPB employed may vary slightly depending on the specific radiology position in question, the current instruments utilized to assess RPB among medical professionals lack standardization and are often evaluated concurrently with knowledge and attitudes toward radiation protection [27–29]. Meanwhile, a further limitation of these studies is that the measurement instruments used were not sufficiently reliable. Indeed, the majority of these instruments were designed by the researchers themselves and were not subjected to any reliability testing [27,28].

The current RPB of medical personnel is inadequate. In some countries, it is estimated that 60~90% of MWs engaged in radiation-related diagnostic and treatment work utilize lead aprons and lead suits for radiation protection [30–32], but only 10~40% of them wear lead gloves and glasses [18,29,33]. Less than 5% of those wear lead hats and even fewer people wear all their radiation protection equipment. The emphasis is placed on the protection of the torso from radiation, with less attention given to the safeguarding of the eyes, head, or other organs. Furthermore, 30~70% of MWs utilize dosimeters at their place of work to monitor and record their radiation doses [18,31,33]. Although standards of radiation protection vary from country to country, the general level of RPB needs improvement.

There are a multitude of factors that contribute to the RPB of MWs, such as medical staff title, position, occupation, education, length of service, fertility, age, hospital level, perception, attitude, and job satisfaction. . . Nevertheless, despite the multitude of potential influences, the sole behavioral intervention for radiation protection currently in existence is a single training session on radiation protection [34,35]. This approach is predicated on the premise that RPB may be indirectly influenced by modifying perceptions or attitudes toward radiation protection. However, a single measure cannot be relied upon as the sole means of enhancing RPB among medical personnel.

## Conclusion

The radiation protection behaviors of medical workers must be clearly defined, measurement tools be universally and reliably implemented, and interventions be diversified and targeted to improve radiation protection behaviors and promote their occupational health of. All in all, a more profound and comprehensive grasp of the extant research on the radiation protection behavior of medical personnel can facilitate the generation of novel research hypotheses that can serve as a foundation for future research endeavors in this domain.

## Supporting information

**S1 Checklist. The PRISMA-ScR checklist.**
(DOCX)

**S2 Checklist. The PRISMA-P 2015 checklist.**
(DOC)

**S1 Table. The initial search strategy for PubMed.**
(DOCX)

## Author Contributions

**Conceptualization:** Xun Liu, Yaqing Liu.

**Investigation:** Xun Liu, Yaqing Liu, Pengyan Xiong.

**Methodology:** Xun Liu, Yaqing Liu, Pengyan Xiong, Sixuan Guo, Lei Zhang.

**Resources:** Sixuan Guo.

**Supervision:** Li Liao.

**Writing – original draft:** Xun Liu, Yaqing Liu.

**Writing – review & editing:** Xun Liu, Yaqing Liu, Pengyan Xiong, Sixuan Guo, Lei Zhang, Li Liao.

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
