## [Decision Letter · Decision Letter 0]

11 Jul 2024

PONE-D-24-19397The radiation protection behavior of medical workers: A scoping review protocolPLOS ONE

Dear Dr. Liao,

Thank you for submitting your manuscript to PLOS ONE. After careful consideration, we feel that it has merit but does not fully meet PLOS ONE’s publication criteria as it currently stands. Therefore, we invite you to submit a revised version of the manuscript that addresses the points raised during the review process.

We look forward to receiving your revised manuscript.

Kind regards,

D.S. Prabakaran

Guest Editor

PLOS ONE

Journal Requirements:

2 Please review your reference list to ensure that it is complete and correct. If you have cited papers that have been retracted, please include the rationale for doing so in the manuscript text, or remove these references and replace them with relevant current references. Any changes to the reference list should be mentioned in the rebuttal letter that accompanies your revised manuscript. If you need to cite a retracted article, indicate the article’s retracted status in the References list and also include a citation and full reference for the retraction notice.

Reviewers' comments:

Reviewer's Responses to Questions

**Comments to the Author**

1. Does the manuscript provide a valid rationale for the proposed study, with clearly identified and justified research questions?

Reviewer #1: Yes

Reviewer #2: Yes

2. Is the protocol technically sound and planned in a manner that will lead to a meaningful outcome and allow testing the stated hypotheses?

Reviewer #1: Yes

Reviewer #2: Yes

3. Is the methodology feasible and described in sufficient detail to allow the work to be replicable?

Reviewer #1: Yes

Reviewer #2: Yes

4. Have the authors described where all data underlying the findings will be made available when the study is complete?

Reviewer #1: Yes

Reviewer #2: Yes

5. Is the manuscript presented in an intelligible fashion and written in standard English?

Reviewer #1: Yes

Reviewer #2: Yes

6. Review Comments to the Author

You may also provide optional suggestions and comments to authors that they might find helpful in planning their study.

Reviewer #1: The study entitled “The radiation protection behavior of medical workers: A scoping review protocol” This study collects information about Definition of radiation protection behavior, including aspects such as current practices of radiation protection in medical settings and the instruments used for measurement. The study aims to examine the current research status of radiation protection behaviours. To identify research gaps and guide future research directions.

This manuscript well organised and it useful for medical field and their workers, However, some suggestions can be considered for minor revision to get published in the journal.

1) Kindly check the Kindly check the grammar and English throughout the manuscript.

2) Add some points in the discussion part

3) Add some recent reference

Reviewer #2: This review explores the current state of research to identify research gaps, guide future research, and promote radiation protection behavior and research directions. However, this study is interesting, and the authors should address the following comments:

1. The authors have not included the number of patients or participants (n =?) in Figure 1. please explain

2. The authors kindly include the study's conclusion.

3. In the discussion section, the authors should discuss this study more in relation to previous studies (please add references).

7. PLOS authors have the option to publish the peer review history of their article (what does this mean?). If published, this will include your full peer review and any attached files.

Reviewer #1: No

Reviewer #2: No

---

## [Author Response · Author response to Decision Letter 0]

24 Jul 2024

Reviewer#1:

1)Kindly check the Kindly check the grammar and English throughout the manuscript.

We were really sorry for our careless mistakes. Thanks for your reminder. We have tried our best to revise the mistake of grammar and English in the revised manuscript.

2)Add some points in the discussion part

As suggested by the reviewer, we have added more points in the discussion part. (P5 Line32-40, P6 Line1-26 marked in yellow in the revised paper)

3)Add some recent reference

We have replaced some old references and added some recent references. (P9-P10 marked in yellow in the revised paper)

Reviewer#2:

1)The authors have not included the number of patients or participants (n =?) in Figure 1. Please explain

We are sorry for this careless mistake, and we have completed the number in Figure 1. in the revised paper.

2)The authors kindly include the study's conclusion.

As suggested by the reviewer, we have added the study's conclusion. (P6 Line27-34 marked in yellow in the revised paper)

3)In the discussion section, the authors should discuss this study more in relation to previous studies (please add references).

We have discussed this study more in relation to previous studies and added more references. (P5 Line32-40, P6 Line1-26 marked in yellow in the revised paper)

---

## [Editor Report · Decision Letter 1]

25 Jul 2024

The radiation protection behavior of medical workers: A scoping review protocol

PONE-D-24-19397R1

Dear Dr. Liao,

We’re pleased to inform you that your manuscript has been judged scientifically suitable for publication and will be formally accepted for publication once it meets all outstanding technical requirements.

Kind regards,

D.S. Prabakaran

Guest Editor

PLOS ONE
---

## [Editor Report · Acceptance letter]

29 Jul 2024

PONE-D-24-19397R1 

PLOS ONE

Dear Dr. Liao, 

I'm pleased to inform you that your manuscript has been deemed suitable for publication in PLOS ONE. Congratulations! Your manuscript is now being handed over to our production team.

Kind regards, 

on behalf of

Dr. D.S. Prabakaran 

Guest Editor

PLOS ONE